# Photonic hyperthermia of malignant peripheral nerve sheath tumors at the third near-infrared biowindow

**Yihui Gu[1†], Zhichao Wang[1†], Chengjiang Wei[1†], Yuehua Li[1], Wei Feng[2], Wei Wang[1], Meiqi Chang[3]\*, Yu Chen[2]\*, Qingfeng Li[1]\***

[1]Department of Plastic and Reconstructive Surgery, Shanghai Ninth People's Hospital, Shanghai Jiao Tong University School of Medicine, Shanghai, China; [2]School of Life Sciences, Shanghai University, Shanghai, China; [3]State Key Laboratory of High Performance Ceramics and Superfine Microstructure, Shanghai Institute of Ceramics, Chinese Academy of Sciences, Shanghai, China

**\*For correspondence:**
changmeiqi@vip.sina.com (MC);
chenyuedu@shu.edu.cn (YC);
dr.liqingfeng@shsmu.edu.cn (QL)

[†]These authors contributed equally to this work

**Competing interest:** The authors declare that no competing interests exist.

## Abstract

**Background:** Malignant peripheral nerve sheath tumors (MPNSTs) are aggressive sarcomas that typically carry a dismal prognosis. Given the insensitivity of these tumors to traditional chemotherapy and the absence of effective targeted drugs, new therapeutic strategies are urgently needed. Photothermal therapy (PTT) including near-infrared laser at the third biowindow (NIR-III) has demonstrated significant potential in cancer theranostics due to its minimally invasive nature and excellent therapeutic outcomes. However, the passive utilization of photothermal agents (PTAs) with poor target specificity and biocompatibility substantially hinders the clinical translation and application of this method.

**Methods:** We evaluated the efficiency, safety, and underlying mechanisms of NIR-III without PTAs in the treatment of MPNSTs. The photothermal performance and tissue penetration capability of the NIR-III laser were evaluated in human MPNST cell lines using CCK-8, Calcein-AM and propidium iodide (PI) staining, and Annexin V-FITC/PI assays. The tumor xenografted mice model was used for evaluating the efficacy and biosafety of NIR-III photothermal ablation. Finally, the underlying mechanisms of NIR-III treatment, explored by whole-transcriptome sequencing, are further verified by RT-qPCR.

**Results:** We found that although the NIR-III photothermal treatment efficiency varied among individuals, which was possibly influenced by different endoplasmic reticulum stress responses, the expected antineoplastic effect was ultimately achieved after adjustment of the power density and radiation duration.

**Conclusions:** The present study provides an intriguing noninvasive therapy for MPNSTs that accelerates the clinical translation of PTT while avoiding the biocompatibility issues arising from PTAs.

**Funding:** This work was supported by grants from National Natural Science Foundation of China (82102344; 82172228); Shanghai Rising Star Program supported by Science and Technology Commission of Shanghai Municipality (20QA1405600); Natural Science Foundation of Shanghai (22ZR1422300); Science and Technology Commission of Shanghai Municipality (19JC1413) ; "Chenguang Program" supported by Shanghai Education Development Foundation (SHEDF) (19CG18); Shanghai Municipal Key Clinical Specialty (shslczdzk00901); Innovative research team of high-level local universities in Shanghai (SSMU-ZDCX20180700).

## Editor's evaluation

Li et al. report on their effort in NIR-III laser-based photothermal therapy for MPNSTs treatment. Importantly, the biosafety issues of photothermal agents could be circumvented through the introduction of NIR-III laser with relatively high penetration depths and low optical scattering effects. in vitro and in vivo results corroborated the relevant conclusions. This work features high novelty on photonic tumor therapy.

## Introduction

Malignant peripheral nerve sheath tumors (MPNSTs) are aggressive soft-tissue sarcomas that arise within peripheral nerves. MPNSTs, which affect approximately 10% of patients with neurofibromatosis type 1 (NF1) and cause severe organ damage, are associated with high morbidity (*Mowery and Clayburgh, 2019*; *Widemann, 2009*). Chemotherapy and radiotherapy provide only limited benefits to MPNST patients, and there is no targeted therapy approved for clinical use (*Vitolo et al., 2019*; *van Noesel et al., 2019*; *Kroep et al., 2011*). The key to improving the prognosis of MPNST patients is extended resection with negative surgical margins, which still faces problems such as the difficulty of surgery, high risk of local recurrence, and distant metastasis (*Widemann and Italiano, 2018*). There is thus a very large unmet medical need to develop efficient therapeutic strategies for MPNSTs.

Photothermal therapy (PTT), a minimally invasive, highly specific, and temporally and spatially selective local treatment modality, has gained increasing attention in the field of cancer theranostics, including for the treatment of MPNSTs (*Huang and Lovell, 2017*). Traditional PTT contains two elements: photothermal agents (PTAs) and lasers. PTAs can be grouped into inorganic and organic materials. Inorganic PTAs include noble metal materials (e.g., Au, Pd, Pt, etc.) (*Tang et al., 2014*; *Shi et al., 2016a*), metal chalcogenide materials (e.g., $Ag_2S$, CuS, $Cu_{2-x}Se$, etc.) (*Sheng et al., 2018*), carbon-based materials (e.g., carbon nanotubes, carbon dots, and graphene) (*Xu et al., 2018*), and graphene analogs (two-dimensional transition metal carbides/nitrides [MXenes], two-dimensional monoelemental materials [Xenes], hexagonal boron nitride, carbon nitride, and transition metal dichalcogenides) (*Luo et al., 2018*; *Thapa et al., 2016*). However, inorganic PTAs have intrinsic issues such as poor biodegradability, disappointing processability, and apparent cytotoxicity. Organic PTAs include mainly small molecules (e.g., cyanine, porphyrin, phthalocyanine, croconaine, boron dipyrromethene, etc.) (*Bhattarai and Dai, 2017*; *Cao et al., 2017*; *Pan et al., 2019*) and semiconducting polymer-based nanoparticles (e.g., polyaniline, polypyrrole, etc.) (*Xia et al., 2017*; *Yang et al., 2018*; *Ma et al., 2019*). Although the advantageous properties of organic agents include distinct biodegradability and biocompatibility, optimization of photothermal conversion efficiency and photothermal stability is highly challenging (*Liu et al., 2019*). Therefore, both organic and inorganic PTAs significantly hinder the clinical application of PTT (*Yuan et al., 2013*).

Near-infrared (NIR) lasers, which represent a promising type of PTT-related laser, play significant roles in addressing the demand for superficial tumor therapy due to their biowindow characteristics, which include a lower background signal and higher tissue penetration depth than those of other biowindows (*Lin et al., 2017*). NIR light can be divided into four distinct regions according to wavelength: NIR-I (700–950 nm), NIR-II (1000–1350 nm), NIR-III (1600–1870 nm), and NIR-IV (2100–2300 nm) (*Sordillo et al., 2017*). The latest research on light-responsive wavelengths for PTT has focused mainly on NIR-I and NIR-II biowindows in combination with NIR lasers and wavelength-matched PTAs (*Guo et al., 2018*; *Feng et al., 2019*). The evidence regarding NIR-III lasers remains comparatively limited in the field of PTT. Compared with NIR-I and NIR-II lasers, NIR-III lasers possess relatively high penetration depths and low optical scattering effects (*Horton et al., 2013*). The total attenuation lengths of human prostate and breast cancer have been revealed to be higher in the NIR-III optical windows than in the other NIR regions, confirming the potential for application of NIR-III lasers in tumor therapy (*Choi et al., 2016*).

The majority of MPNSTs are located on the body surface including head, neck, and extremity (*Le Guellec et al., 2016*). Therefore, it could be boldly speculated that NIR-III laser would effectively induce photothermal ablation of MPNSTs due to their surface layout characteristics. *Sordillo et al., 2017* previously demonstrated that NIR-III treatment appears to involve a better wavelength range for tissue containing high collagen than conventional NIR-I treatment. In addition, a previous genetic profiling study has revealed that collagen expression is significantly upregulated in MPNSTs (*Karube et al., 2006*). Therefore, the photothermal effects of NIR-III could be more concentrated in MPNSTs

than in adjacent normal skin and soft tissue. Moreover, the safety of NIR-III has been widely demonstrated in various medical applications, including optical imaging (*Shi et al., 2016b*), eye disease treatment (*Sakimoto et al., 2006*), and even cosmetic treatments (*Adatto et al., 2017*). In summary, the potential therapeutic effectiveness and desirable safety of NIR-III treatment will likely accelerate its clinical application compared with that of traditional photothermal theranostic methods. Although the application of NIR-III for PTT has not yet been demonstrated in MPNSTs, the existing data further strengthen our expectation that NIR-III laser can be used for PTT of tumors without the utilization of PTAs.

In this work, with the aim of reducing the pain of MPNST patients (*Figure 1a–b*) and avoiding the intrinsic drawbacks of PTAs and the aforementioned critical problems, we performed PTT using a 1650 nm NIR-III laser in a simple manner without the addition of PTAs (*Figure 1—figure supplement 1*). The relatively high collagen content of MPNSTs ensure the efficiency of photothermal tumor ablation and make the NIR-III laser a suitable and potent therapeutic modality that has potential for use in clinical applications. In principle, the photothermal conversion of the NIR-III laser is attributed to the thermal molecular movement of the irradiated object after laser irradiation. In addition, the power safety limit for the NIR-III laser is higher than that for NIR-I and NIR-II lasers, which in turn leads to efficient and safe tumor ablation (*Figure 1c–d*). Both in vitro and in vivo results indicate that the photothermal conversion ability of the NIR-III laser is enough to rapidly ablate cancer cells, which affirms the antineoplastic effect of NIR-III PTT. This PTAs-free PTT strategy possesses unique advantages for the development of clinical application and the improvement of patient prognosis due to the excellent biocompatibility and predominant photothermal conversion efficiency of NIR-III laser.

# Materials and methods
## Materials and reagents

Dulbecco's modified Eagle's medium (DMEM), fetal bovine serum (FBS), and penicillin/streptomycin were purchased from Gibco, USA. A Cell Counting Kit-8 (CCK-8) and Calcein/PI Cell Viability/Cytotoxicity Assay Kit were purchased from Dojindo, JP. A FITC-Annexin V Apoptosis Detection Kit was purchased from BD Biosciences, USA.

## Photothermal performance of the NIR-III laser

The laser was customized by Shanghai Xilong Optoelectronics Technology Co., Ltd. The characteristics of the NIR-III laser were tested and verified as described in *Supplementary file 1*. To investigate the photothermal performance of the NIR-III laser, deionized water was irradiated by the NIR-III laser at different power densities (0.25, 0.5, 0.75, 1, 1.25, and 1.5 W cm$^{-2}$) for 15 min. The surfaces of chicken breasts with different thicknesses (0, 2, 4, 6, 8, and 10 mm) was irradiated by the NIR-III laser at different power densities (0.25, 0.5, 0.75, 1, 1.25, and 1.5 W cm$^{-2}$) for 10 min. Fat of different thicknesses (0, 2, 4, 6, 8, and 10 mm) was also irradiated by the NIR-III laser at different power densities (0.25, 0.5, 0.75, 1, 1.25, and 1.5 W cm$^{-2}$) for 10 min. Temperature changes were monitored by an infrared thermal camera, and the relevant data were analyzed with AnalyzIR software. In addition, to compare the photothermal performance of lasers with different wavelengths, deionized water was also irradiated by 808, 980, and 1064 nm lasers with different power densities (0.25, 0.5, 0.75, 1, 1.25, and 1.5 W cm$^{-2}$) for 10 min.

## Cell lines and cell culture

Two MPNST cell lines (STS26T and S462) were generously donated by Dr Vincent Keng. S462 was derived from an NF1-MPNST patient, whereas STS26T was derived from a sporadic MPNST patient. The two MPNST cell lines were maintained in high-glucose DMEM with 10% FBS and 1% penicillin/streptomycin in a 37°C, 5% CO$_2$ incubator providing a humidified atmosphere. The culture medium was replaced every 3 days, and the cells were passaged using 0.25% trypsin until they approached 80% confluence. All cell lines were tested mycoplasma negative every 3 months. Verification of cell lines was confirmed by Short Tandem Repeat DNA profiling (Applied Biological Materials Inc, Canada).

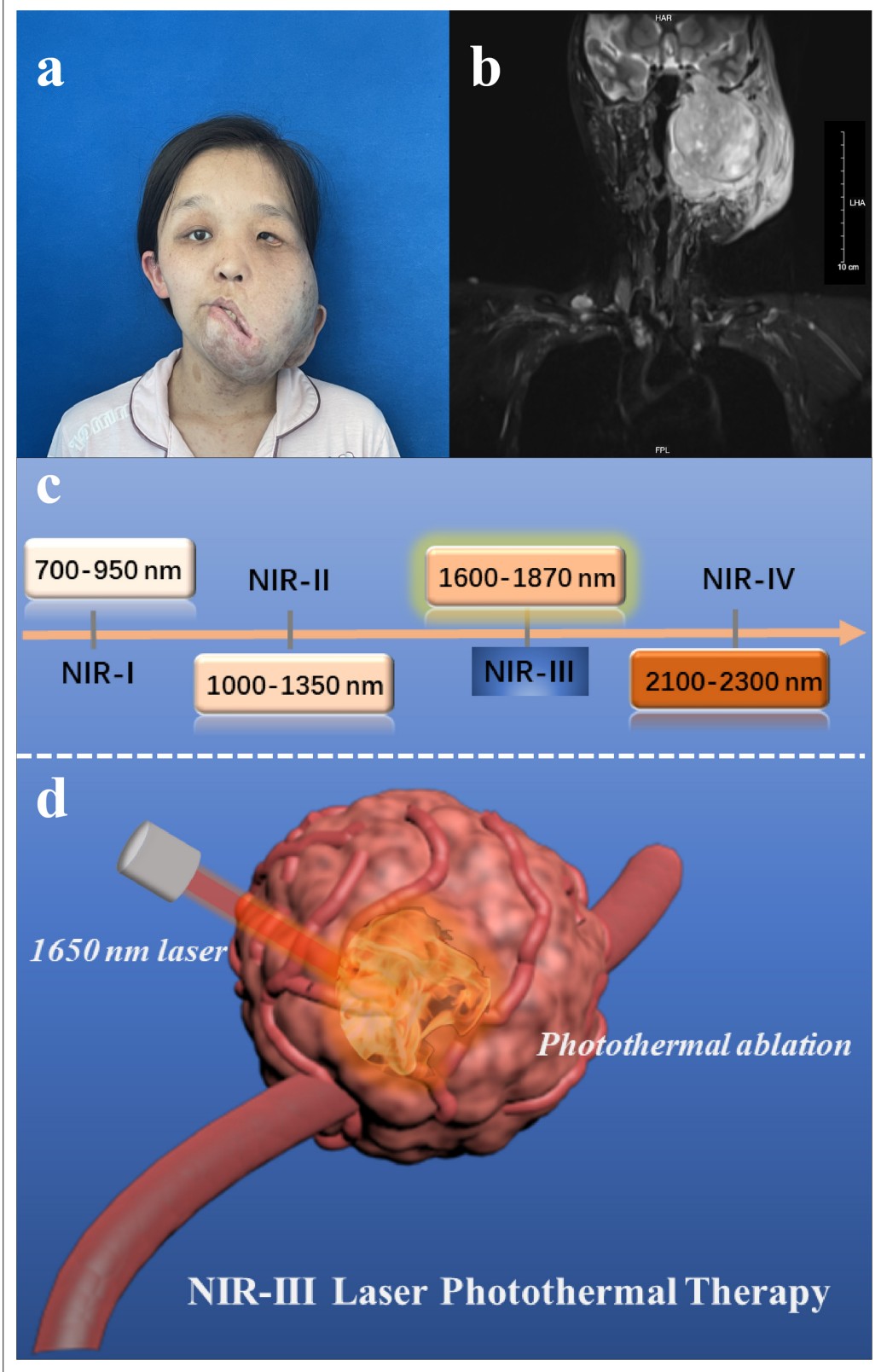

**Figure 1.** Disease models and therapeutic illustrations.
(**a**) Photograph and (**b**) magnetic resonance images of malignant peripheral nerve sheath tumor (MPNST) patient. (**c**) Classification of four near-infrared (NIR) wavelength regions for biomedical applications. (**d**) Schematic

*Figure 1 continued on next page*

*Figure 1 continued*

illustration of NIR-III laser (1650 nm) tumor photothermal therapy (PTT) through photothermal ablation without the addition of photothermal agents (PTAs).

The online version of this article includes the following figure supplement(s) for figure 1:

**Figure supplement 1.** Schematic diagram of the experimental process.

## Cell viability assays

A CCK-8 cell viability assay was implemented to assess the photothermal ablation effect and penetration capability of the NIR-III laser. MPNST cells (STS26T and S462) were seeded at a density of 10,000 cells per well in 96-well cell culture plates and cultured overnight for adherence. The adherent cells were irradiated by the NIR-III laser at different power densities (0.25, 0.5, 0.75, 1, 1.25, 1.5 W $cm^{-2}$) for 5 and 10 min. The cells were irradiated by the NIR-III laser at a power density of 0.5 or 1 W $cm^{-2}$ for 5 min with chicken skin of different thicknesses (2, 4, and 6 mm) covered on the culture plates. After the different irradiation protocols, the cells were incubated at 37°C under 5% $CO_2$ for 30 min. Subsequently, the culture medium was replaced with 100 µl of DMEM containing 10 µl of CCK-8 reagent per well, and the cells were incubated for 2 hr at 37°C. The absorbance was recorded at 450 nm.

## Confocal laser scanning microscopy

Confocal laser scanning microscopy (CLSM) imaging was also performed to further evaluate the photoinduced damage caused by the NIR-III laser in MPNST cells. STS26T and S462 cells were seeded on coverslips in 96-well cell culture plates and cultured overnight for adherence. After the different treatments mentioned above, the cells were rinsed with phosphate-buffered saline (PBS) three times and stained with calcein-AM and PI in PBS for 15 min in a $CO_2$ incubator. After 15 min of staining, the cells were washed three times with PBS and then observed and recorded by CLSM (Ex: 488 nm, Em: 515 nm for live cells; Ex: 535 nm, Em: 617 nm for dead cells).

## Flow cytometry

An Annexin V-FITC/propidium iodide (PI) assay was employed to detect apoptotic and necrotic cells. MPNST cells (STS26T and S462) were seeded in 96-well cell culture plates ($5×10^4$ cells suspended in 100 µl of culture medium per well) and irradiated by the NIR-III laser for different durations (2, 5, and 10 min) at different power densities (0.5 and 1 W $cm^{-2}$). The NIR-III laser was focused on the middle of each plate. Similar experiments were also conducted in 24-well cell culture plates ($2×10^5$ cells suspended in 400 µl of culture medium per well). After photothermal treatment with the NIR-III laser, $1×10^5$ MPNST cells (STS26T and S462) were resuspended in 100 µl of binding buffer, and 5 µl of Annexin V-FITC and 5 µl of PI were added sequentially. The stained cells were incubated at room temperature in the dark for 15 min and then analyzed using a flow cytometer (Beckman Coulter, Shanghai) equipped with CytExpert software. The cells were divided into viable cells, early apoptotic cells, late apoptotic or dead cells, and necrotic cells.

## Animal xenograft tumor model

Four-week-old male immunodeficient athymic nude mice were purchased from Shanghai Model Organisms Co., Ltd. (Shanghai, China). All animals received humane care in compliance with the guidelines outlined in the Guide for the Care and Use of Laboratory Animals. All procedures were performed in accordance with the guidelines approved by the Shanghai Medical Experimental Animal Care Commission (IACUC: 2019-0008). The MPNST cell line STS26T and S462 was used to establish a cell line-based xenograft model. A total of $5×10^6$ STS26T cells or $1×10^7$ S462 cells suspended in 100 µl of PBS were engrafted subcutaneously on the back of each mouse.

## Tumor temperature monitoring during laser irradiation

To investigate the in vivo photothermal performance of the NIR-III laser, the temperatures of STS26T and S462 xenografts irradiated at two different power densities (0.5 and 1 W cm$^{-2}$) were monitored after different irradiation durations (0, 2, 4, 6, 8, and 10 min). The temperature changes were monitored by an infrared thermal camera, and the relevant data were analyzed with AnalyzIR software.

## Deep-tissue PTT

The tissue penetration ability of the NIR-III laser for photothermal ablation was assessed. When the tumor volume reached approximately 1500 mm$^3$, the STS26T tumor-bearing mice were irradiated at two different power densities (0.5 and 1 W cm$^{-2}$) for 5 min. The tumors were harvested 10 hr post treatment and sectioned into slices at different depths (1.5, 3, 4.5, 6, 7.5 mm). The sections were stained with H&E or with antibodies against Cleaved-Caspase 3 and Ki-67 for further histological analysis.

## In vivo PTT

The efficacy and safety of NIR-III PTT were evaluated simultaneously. When the average tumor volume reached 100 mm$^3$, the STS26T tumor-bearing mice were divided into three groups: the control group, the low-power density group (0.5 W cm$^{-2}$), and the high-power density group (1 W cm$^{-2}$). All mice were anesthetized before receiving NIR-III laser irradiation for 5 min. After the treatment, a proportion of the mice in three groups were euthanized and the tumors were harvested for histological analysis. The rest of the mice were closely observed for the changes in tumors and health over the next 15 days. The tumor sizes were measured by a digital caliper every 2 days, and the tumor volumes were calculated with the equation $0.5 \times L \times W^2$, where L is the longest diameter and W is the width. The mice were euthanized once the tumor volume reached 1500 mm$^3$ or the maximum diameter of the tumor reached 2 cm. The body weights were also recorded every 2 days to reflect the health of the mice. After close observation for 15 days, all tumors were excised and weighed, and then the major organs were dissected and processed by using H&E staining for histological evaluation.

## Histology and immunohistochemistry

Tissue processing, embedding, and sectioning were performed by Biosci Biotechnology. The paraffin sections were stained with H&E, TUNEL, a Ki-67 antibody and a Cleaved-Caspase 3 antibody for histological analysis according to standard protocols with minor modifications due to antibody optimization. Commercially available antibodies were utilized for immunohistochemical staining of Ki-67 (1:400, 9449, Cell Signaling Technology) and Cleaved-Caspase 3 (1:500, 9664, Cell Signaling Technology). A TUNEL Apoptosis Detection Kit (FITC) (40306ES20, Yeasen Biotech) was used for TUNEL staining.

## RNA-seq analysis

To explore the transcriptional alterations of MPNST after the NIR-III treatment, RNA-seq analysis was conducted in STS26T and S462 xenografts. After the successful establishment of the xenografts, the tumor-bearing mice were anesthetized and irradiated at two different power densities (0.5 and 1 W cm$^{-2}$) for 5 min. Immediately after the treatment, the mice in two treatment groups and control group were euthanized and the tumors were harvested for RNA-seq analysis (n=1 for each group) and RT-qPCR analysis. Total RNA of the tumors was extracted using TRIzol reagent (Invitrogen) according to the manufacturer's instructions. RNA-seq data were generated using an Illumina HiSeq platform. A reference genome and gene model annotation files were downloaded directly from the genome website. STAR (v2.5.1b) was used to build an index of the reference genome and to align the paired-end clean reads to the reference genome. STAR uses the Maximal Mappable Prefix (MMP) method, which can generate precise mapping results for junction reads. HTSeq v0.6.0 was used to count the reads numbers mapped to each gene. And then FPKM (the fragments per kilobase of transcript per million mapped reads) of each gene was calculated based on the length of the gene and

reads count mapped to this gene. Prior to differential gene expression analysis, for each sequenced library, the read counts were adjusted by the edgeR program package with one scaling normalization factor. Differential expression analysis between two conditions was performed using the edgeR R package (v3.12.1). The p-values were adjusted using the Benjamini & Hochberg method. padj <0.05, |LFC|>1, and FPKM >1 in at least one sample were set as the thresholds for significant differential expression.

## Enrichment analysis

GO enrichment analysis was performed with the clusterProfiler R package with correction for gene length bias. GO terms with corrected p-values less than 0.05 were considered significantly enriched. The KEGG is a database resource for understanding the high-level functions and utilities of biological systems, such as cells, organisms, and ecosystems, from molecular-level information, especially large-scale molecular datasets generated by genome sequencing and other high-throughput experimental techniques (http://www.genome.jp/kegg/). We used the clusterProfiler R package to test the statistical enrichment in KEGG pathways.

## Statistical analysis

Statistical analysis was conducted using Prism 8.0 (GraphPad Software, San Diego, CA). The statistical analyses included the chi-square test, Student's t-test, and one-way ANOVA, as appropriate. p-values <0.05 were considered to indicate statistical significance, and asterisks (*) are used to indicate significant differences between two specified groups. ** indicates a p-value <0.01, while *** indicates a p-value <0.001. p-Values >0.05 qualified as not statistically significant.

## Results

The 1650 nm NIR-III laser used in this work was composed of laser diodes, a collimator lens, a photodetector, a spectroscope, focusing lenses, and an optical fiber interface (*Figure 2a*). The relevant laser parameters are listed in *Supplementary file 1*. The photothermal effectiveness of the NIR-III laser was initially evaluated by measuring the temperature changes of an aqueous solution. A schematic diagram and the equipment are shown in *Figure 2b*. The temperature of the aqueous solution increased gradually with increasing irradiation time and power density (*Figure 2c–d*). Notably, the 1.5 W cm$^{-2}$ laser caused an ~40°C temperature increase within 10 min.

To further distinguish the photothermal efficiencies of lasers in different biowindows, the temperature changes of aqueous solutions irradiated by NIR-I (808 nm) and NIR-II (980 and 1064 nm) lasers at different power densities were also recorded. For the NIR-I group, adjustment of the laser power had a negligible effect on the change in aqueous solution temperature (*Figure 2—figure supplement 1*). In contrast, in the NIR-II group, an increase in power density elevated the aqueous solution temperature to a certain extent. However, the maximum temperature difference was controlled within 15°C regardless of the duration (*Figure 2—figure supplements 2 and 3*). The results above confirm the excellent photothermal performance of the NIR-III laser in comparison to traditional NIR-I and NIR-II lasers.

The NIR-III laser must penetrate skin and subcutaneous fat before it can produce a tumor ablation effect. Therefore, we further evaluated the tissue penetration capability of the NIR-III laser using different biological tissue models (*Figure 3a*). First, the dependence of the temperature increment on power density was evaluated in chicken breast models (*Figure 3b and c*). The temperature of the chicken breast surface rapidly increased from 27.2°C to 76.7°C within 10 min under NIR-III laser irradiation (1.5 W cm$^{-2}$). In addition, the results of photothermal penetration experiments further validated the highly efficient photothermal performance of the NIR-III laser, as shown in *Figure 3d and e*. The difference between the temperature at 0 s and the temperature at 600 s (ΔT) was still over 10°C even when a chicken breast of 10 mm thickness was used to block the radiation. Similar results were obtained in the pig fat penetration experiments, namely, the extent of the temperature change was proportional to the power density and inversely proportional to the fat thickness (*Figure 3f–i*). Notably, temperature range (36.6–91.1°C) varied with laser power and fat thickness. In addition, the temperature alterations at various tissue depths indicated that similar attenuation of photothermal

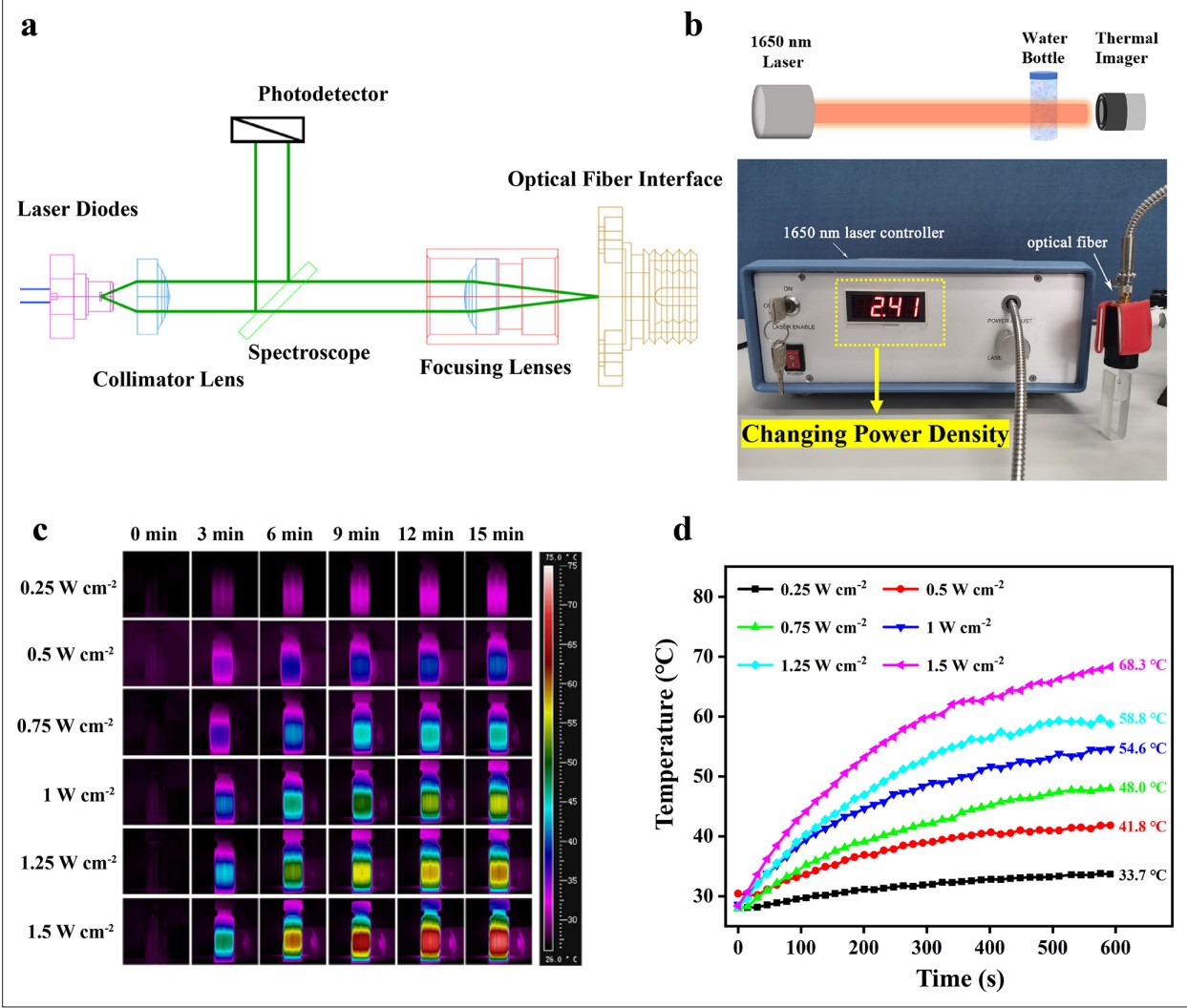

**Figure 2.** Photothermal properties of the NIR-III laser. (**a**) Circuit diagram of the NIR-III laser. (**b**) Schematic diagram and equipment for detection of the photothermal effectiveness of the NIR-III laser. (**c**) Photothermal images of aqueous solutions irradiated for various durations with a 1650 nm laser at different power densities. (**d**) Temperature curves of aqueous solutions irradiated by a 1650 nm laser at different power densities.

The online version of this article includes the following figure supplement(s) for figure 2:

**Figure supplement 1.** Photothermal behaviors of aqueous solutions by an 808 nm laser.

**Figure supplement 2.** Photothermal behaviors of aqueous solutions by an 980 nm laser.

**Figure supplement 3.** Photothermal behaviors of aqueous solutions by an 1064 nm laser.

efficiency could be obtained regardless of whether the biological tissue was chicken breast or pig fat (*Figure 3j*).

Two representative MPNST cell lines, STS26T and S462, were utilized for in vitro experiments. First, the influences of power density and irradiation time on the anticancer efficacy of the NIR-III laser were assessed in STS26T cells (*Figure 4a*). In the 5 min group, the 0.25 W cm$^{-2}$ NIR-III laser induced a 25% reduction in cell viability, while lasers with higher power density all exhibited a strong killing effect. In the 10 min group, NIR-III irradiation caused maximum cell death regardless of the power density. In addition, the tumor killing efficiency of the NIR-III laser at different penetration depths was also evaluated. The NIR-III laser (1 W cm$^{-2}$) demonstrated almost 100% cell killing efficiency after penetrating chicken skin thickness of 4 mm (*Figure 4b*). Similar results were obtained in the S462 cell groups (*Figure 4c and d*). A significant increase in S462 cell viability was observed with a chicken skin thickness of 6 mm. For a more thorough examination, calcein-AM/propidium iodide (PI) assays were implemented to investigate the relationship between power density/irradiation time and cell viability.

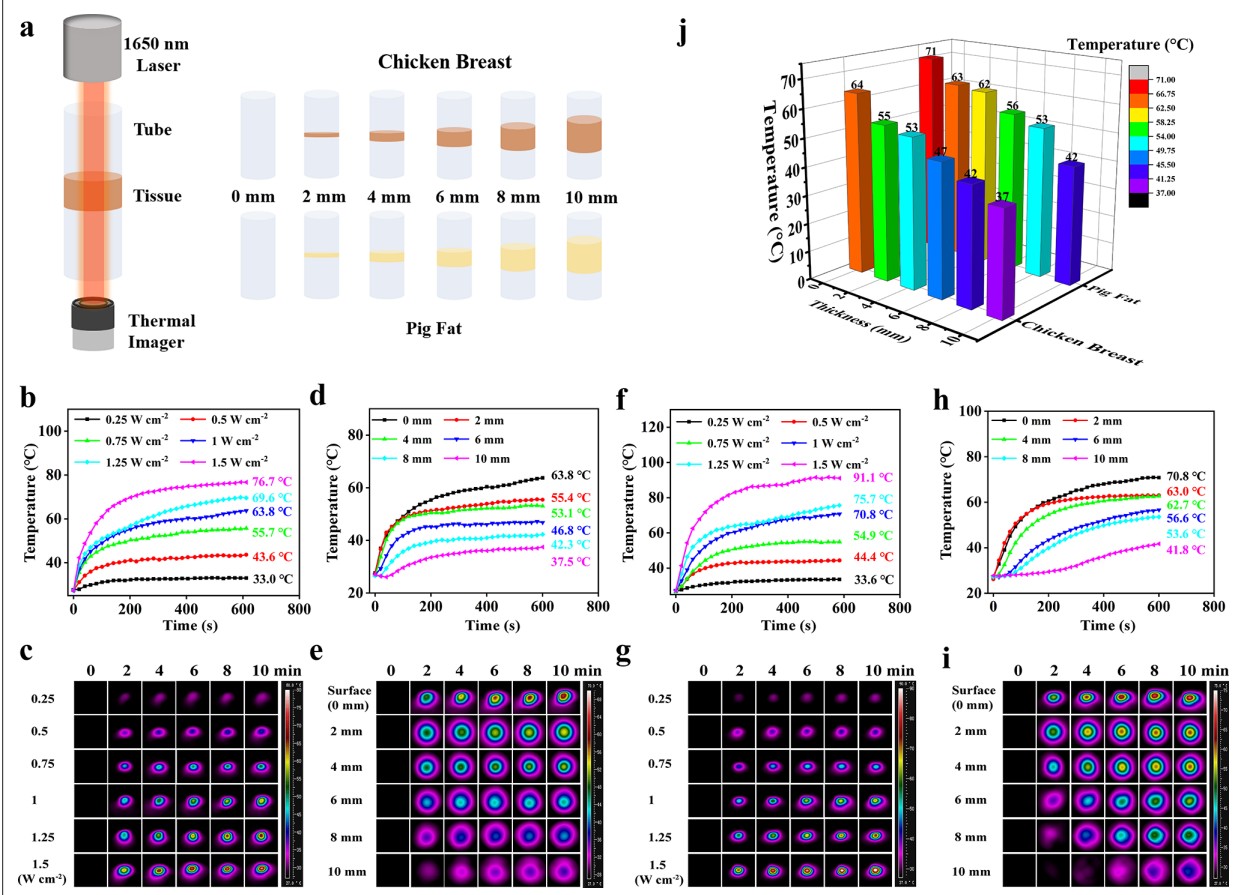

**Figure 3.** Photothermal performance for skin and subcutaneous tissue penetration. (**a**) Schematic diagram for detection of the tissue penetration capability of the NIR-III laser. (**b–c**) Temperature curves (**b**) and photothermal images (**c**) of chicken breast surfaces irradiated by the NIR-III laser at different power densities. (**d–e**) Temperature curves (**d**) and photothermal images (**e**) of chicken breasts with different thicknesses irradiated by the NIR-III laser at 1.0 W cm$^{-2}$. (**f–g**) Temperature curves (**f**) and photothermal images (**g**) of pig fat surfaces irradiated for various durations with the NIR-III laser at different power densities. (**h–i**) Temperature curves (**h**) and photothermal images (**i**) of pig fat with different thicknesses irradiated for various durations with the NIR-III laser at 1.0 W cm$^{-2}$. (**j**) Bar graphs of temperature changes in chicken breast and pig fat of different thicknesses caused by NIR-III laser irradiation for 600 s.

The results demonstrated that significant cell death occurred in NIR-III (0.5 W cm$^{-2}$, 5 min) group, the NIR-III (1 W cm$^{-2}$, 2 min) group, and the NIR-III (1 W cm$^{-2}$, 5 min) groups in both STS26T and S462 cells (*Figure 4e and f*, *Figure 4—figure supplement 1*). In addition, consistent results were obtained through flow cytometric analysis. For the NIR-III (1 W cm$^{-2}$, 2 min), NIR-III (1 W cm$^{-2}$, 5 min), NIR-III (1 W cm$^{-2}$, 10 min), NIR-III (0.5 W cm$^{-2}$, 5 min), and NIR-III (0.5 W cm$^{-2}$, 10 min) groups, the late apoptotic and dead cell populations reached almost 100%, confirming the outstanding photothermal ablation effect of the NIR-III laser (*Figure 4g and h*). To support clinical applications, the optimal power density and irradiation time should be investigated to maximize the tumor ablation efficiency of the NIR-III laser.

STS26T and S462 MPNST xenograft mouse models were constructed to explore the capacity of the NIR-III laser at 0.5 and 1.0 W cm$^{-2}$ to alter temperature and achieve photothermal ablation in vivo, as shown in *Figure 5a–d*. In the STS26T xenograft model (*Figure 5a and b*), the tumor temperature increased from 32.9°C and 33.9°C to 60.2°C and 75.4°C within 10 min of irradiation at the low (ΔT=27.3°C) and high (ΔT=41.5°C) power densities, respectively. Additionally, after adjustment of the laser power density and irradiation time, similar increasing trends were observed in the S462 xenograft group for the low (ΔT=18.6°C) and high (ΔT=30.7°C) power densities (*Figure 5c and d*). The excellent photothermal performance in vivo further affirms the potential of the NIR-III laser.

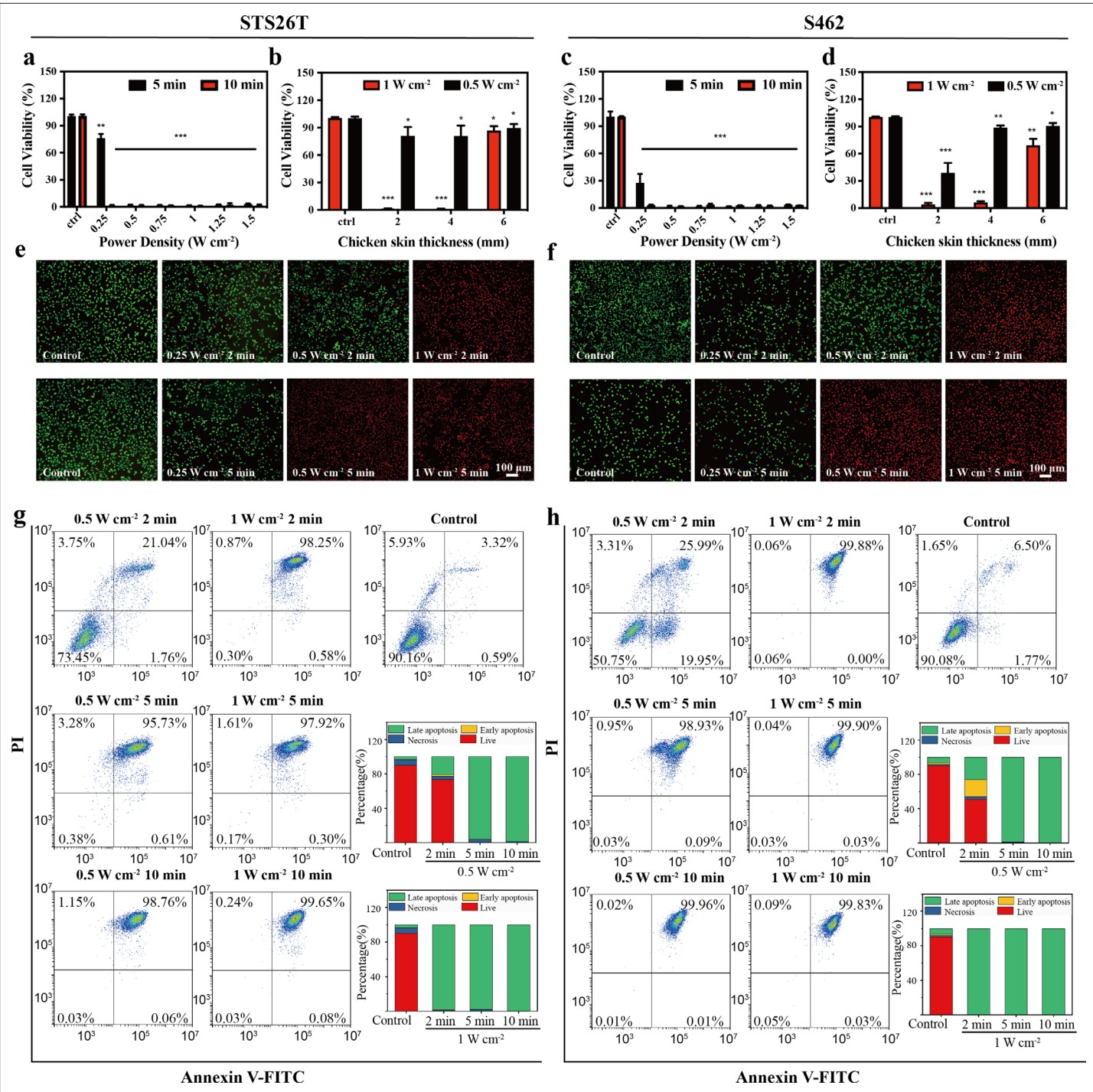

**Figure 4.** In vitro photothermal therapy (PTT) in the NIR-III biowindow. (**a, c**) Relative viabilities of STS26T cells (**a**) and S462 cells (**c**) seeded in 96-well culture plates and subjected to various durations of irradiation with the NIR-III laser at different power densities. (**b, d**) Relative viabilities of STS26T cells (**b**) and S462 cells (**d**) treated with the NIR-III laser (0.5 and 1 W cm$^{-2}$) under different chicken skin thickness. All data are presented as mean ± SD and were analyzed with the unpaired t-test (n=3). (**e–f**) Confocal laser scanning microscopy (CLSM) images of STS26T cells (**e**) and S462 cells (**f**) costained with calcein-AM (green) and propidium iodide (PI) (red) after different treatments. (**g–h**) Flow cytometric analysis using Annexin V-FITC/PI kits in STS26T cells (**g**) and S462 cells (**h**) after different treatments. *p<0.05,**p<0.01, and ***p<0.001.

The online version of this article includes the following figure supplement(s) for figure 4:

**Figure supplement 1.** Cell viability analysis of STS26T cells (**a**) and S462 cells (**b**) using calcein-AM (green) and propidium iodide (PI) (red) staining after different treatments.

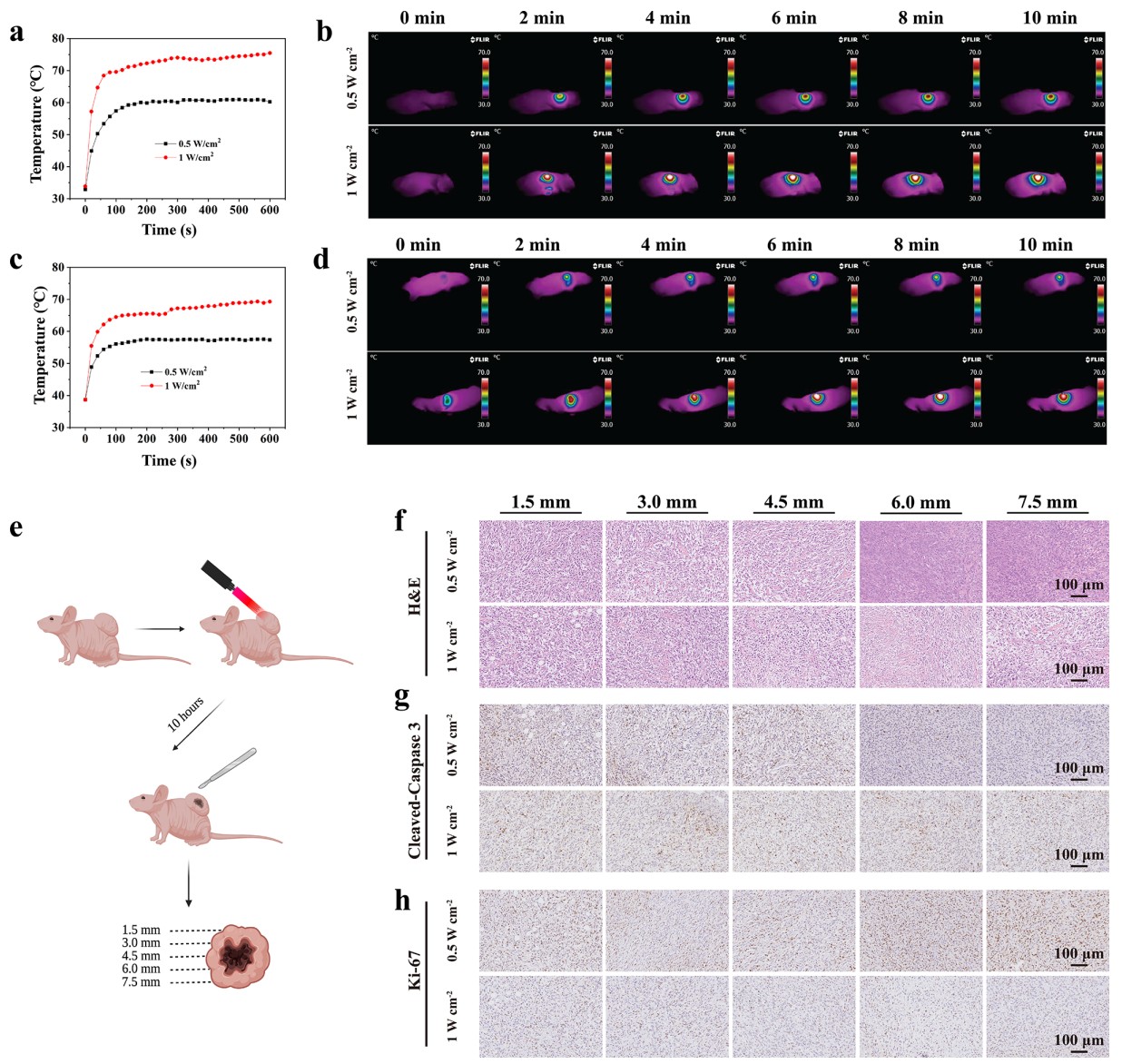

**Figure 5.** In vivo photothermal ablation of malignant peripheral nerve sheath tumors (MPNSTs) in the NIR-III biowindow. (**a–b**) Temperature curves (**a**) and photothermal images (**b**) of STS26T tumor-bearing mice irradiated with the NIR-III laser at different power densities. Temperature curves (**c**) and photothermal images (**d**) of S462 tumor-bearing mice irradiated with the NIR-III laser at different power densities. (**e**) Schematic diagram for detection of the STS26T tumor tissue penetration capability of the NIR-III laser. (**f–h**) Hematoxylin and eosin (H&E) staining for pathological changes (**f**) anti-Cleaved-Caspase 3 immunohistochemical staining for apoptosis (**g**) and anti-Ki-67 immunohistochemical staining for cellular proliferation (**h**) at different depths (1.5, 3, 4.5, 6, and 7.5 mm) of dissected STS26T tumor tissues from the NIR-III (0.5 W cm$^{-2}$, 5 min) group and the NIR-III (1 W cm$^{-2}$, 5 min) group.

The online version of this article includes the following figure supplement(s) for figure 5:

**Figure supplement 1.** The statistical analysis of (**a**) Ki67-positive cells and (**b**) Cleaved-Caspase 3-positive cells after immunohistochemical staining.

To further evaluate the in vivo tissue penetration depth of NIR-III photothermal ablation, STS26T xenograft tissues were dissected, and histological changes, apoptosis, and cell proliferation were observed at different depths (*Figure 5e*). It was revealed that effective photothermal ablation of subcutaneous xenograft tumors could be achieved at a depth of 4.5 mm in the NIR-III (0.5 W cm$^{-2}$, 10 min) group. However, in the NIR-III (1 W cm$^{-2}$, 10 min) group, the effective photothermal ablation depth was over 7.5 mm, as revealed by hematoxylin and eosin (H&E) staining, Ki-67 immunohistochemical staining, and Cleaved-Caspase 3 immunohistochemical staining (*Figure 5f–h*) and the

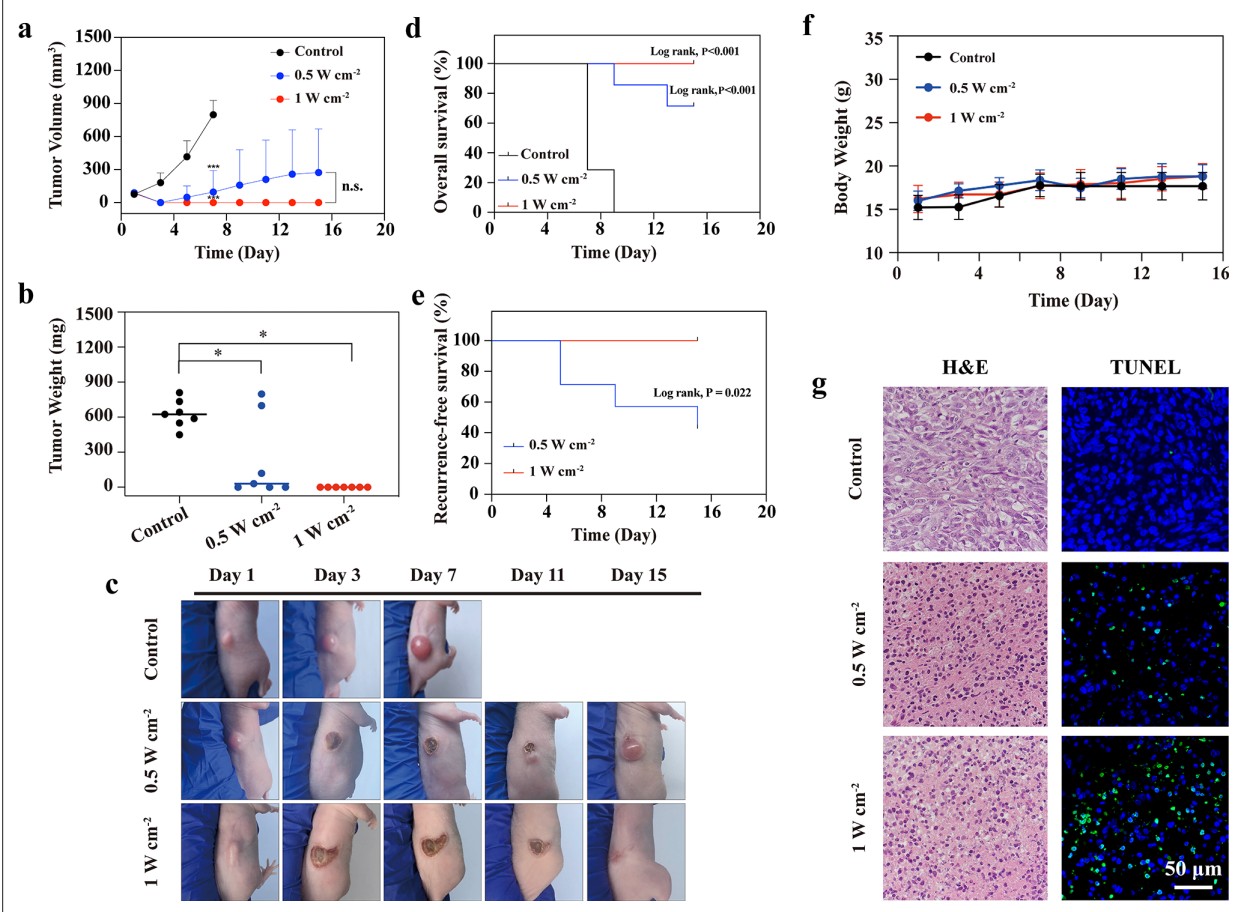

**Figure 6.** In vivo photothermal therapy (PTT) in the NIR-III biowindow. (**a**) Time-dependent tumor growth curves for the control, NIR-III (0.5 W cm$^{-2}$, 5 min), and NIR-III (1 W cm$^{-2}$, 5 min) groups. (**b**) Tumor weights for tumor-bearing nude mice after different treatments. All data are presented as mean ± SD and were analyzed with the unpaired t-test (n=7). 'NS' indicates 'not significant'. (**c**) Photographs of STS26T tumor-bearing mice and their tumor regions at 1, 3, 7, 11, and 15 days after different treatments. (**d–e**) Overall survival (OS) curves (**d**) and recurrence-free survival (RFS) curves (**e**) of mice after various treatments (n=7, p-value, log-rank test). (**f**) Time-dependent body weight curves of nude mice after different treatments (n=7). (**g**) Hematoxylin and eosin (H&E) staining and TdT-mediated dUTP-biotin nick and labeling (TUNEL) staining for pathological changes in tumor tissues from each group. *p<0.05.

The online version of this article includes the following figure supplement(s) for figure 6:

**Figure supplement 1.** The statistical analysis of TdT-mediated dUTP-biotin nick and labeling (TUNEL)-positive cells according to TUNEL staining.

**Figure supplement 2.** Hematoxylin and eosin (H&E) staining of major organs (heart, liver, spleen, lung, and kidney) in the control, NIR-III (0.5 W cm$^{-2}$, 5 min), and NIR-III (1 W cm$^{-2}$, 5 min) groups.

corresponding statistic analysis (*Figure 5—figure supplement 1*). Therefore, it could be concluded that a higher power density is correlated with better therapeutic efficiency and outcomes.

To simultaneously evaluate the efficacy and safety of NIR-III PTT, further in vivo PTT experiments were conducted when the tumor volume reached approximately 100 mm$^3$. The STS26T tumor-bearing mice were divided into three groups: the control group, the NIR-III (0.5 W cm$^{-2}$, 5 min), group and the NIR-III (1 W cm$^{-2}$, 5 min) group. The tumor volume of each mouse was recorded every other day (*Figure 6a*). Two days after photothermal ablation treatment, the tumors in the two treated groups could no longer be observed with the naked eye and had been replaced by black scars. However, four out of seven mice in the NIR-III (0.5 W cm$^{-2}$, 5 min) group had tumor recurrence after 15 days. In comparison, the tumors in the NIR-III (1 W cm$^{-2}$, 5 min) group were completely eradicated (*Figure 6b and c*). In conclusion, the long-term effectiveness evaluation confirmed that the NIR-III (1 W cm$^{-2}$, 5 min) treatment group had the best outcome in terms of survival rate in STS26T tumor-bearing mice (*Figure 6d and e*). Pathological changes revealed by H&E staining and TdT-mediated dUTP-biotin nick and labeling (TUNEL) staining in tumor slices are shown in *Figure 6g*. More necrotic regions were

observed in the NIR-III (1 W cm$^{-2}$, 5 min) group than in the NIR-III (0.5 W cm$^{-2}$, 5 min) group. In addition, apoptosis occurred in the NIR-III (0.5 W cm$^{-2}$, 5 min) group according to TUNEL assays, but the apoptosis in the NIR-III (1 W cm$^{-2}$, 5 min) group was more obvious (*Figure 6—figure supplement 1*).

Furthermore, to test the biosafety of NIR-III treatment, major organs, including the heart, liver, spleen, lungs, and kidneys were collected, and sliced for H&E staining (*Figure 6—figure supplement 2*). No significant pathological toxicity was observed in the two groups treated with PTT, and all of the mice demonstrated negligible weight fluctuations (*Figure 6f*), confirming that NIR-III-induced photothermal ablation had negligible adverse effects.

To explore the underlying mechanisms of the effects of NIR-III treatment, whole-transcriptome sequencing was performed on S462 and STS26T xenograft tumors from the control, NIR-III (0.5 W cm$^{-2}$, 5 min), and NIR-III (1 W cm$^{-2}$, 5 min) groups. In STS26T tumors, significant transcriptional alterations were observed and were further analyzed (*Figure 7a*). Both upregulated and downregulated overlapping mRNAs were found in the NIR-III (0.5 W cm$^{-2}$, 5 min) group and the NIR-III (1 W cm$^{-2}$, 5 min) group compared with control group (*Figure 7b*). According to Kyoto Encyclopedia of Genes and Genomes (KEGG) enrichment analyses, the protein processing in endoplasmic reticulum (ER) pathway (hsa04141) were among the most enriched pathways (*Figure 7c*). Gene Ontology (GO) enrichment analysis of the genes related to the protein processing in ER pathway (hsa04141) was further performed, which revealed that the key biological processes including the ubiquitin-dependent protein catabolic process, the response to unfold protein were enriched, indicating that NIR-III treatment could lead to ER stress in STS26T xenograft tumors (*Figure 7d*).

ER stress is a condition during which a variety of pathological conditions including high temperature may impede the capacity of cells to properly fold secretory and transmembrane proteins in the ER and lead to the accumulation of misfolded proteins. Cells with ER stress can initiate a self-protection mechanism termed the unfolded protein response (UPR) that induces a set of transcriptional and translational events to restore ER homeostasis (*Qi and Chen, 2019*). To explore the specific mechanism, we investigated the interactions between ER pathway-related genes (*Figure 7e*). It could be found that the key transcription factors regulating the UPR (ATF4 and ATF6B) were among the top upregulated mRNAs related to protein processing in ER pathway (*Supplementary file 2*). RT-qPCR analysis also confirmed the significant upregulation of ATF4 and ATF6B in the NIR-III (0.5 W cm$^{-2}$, 5 min) group and the NIR-III (1 W cm$^{-2}$, 5 min) group (*Figure 7f*). It was demonstrated in previous studies that ATF4 and ATF6B contributed to stress relief and survival by inducing the expression of ER chaperones, which have critical functions in facilitating protein folding and assembly. We identified the most significantly upregulated ER chaperones genes P4HB and HSP1A1 and further confirmed the upregulation of these two genes using RT-qPCR (*Figure 7f*).

In addition to ER chaperones, the UPR regulated by ATF4 and ATF6 also upregulates the components of ER-associated degradation (ERAD). Proteins that do not properly fold within a certain time are targeted for ERAD, which efficiently retro-translocates them from the ER into the cytosol for degradation via the ubiquitin-proteasome system. According to KEGG enrichment analyses, the ubiquitin-mediated proteolysis pathway (hsa04120) was also enriched (*Figure 7c*). GO enrichment analyses revealed the related key biological processes including protein polyubiquitination (*Figure 7—figure supplement 1a*). The interactions between ubiquitin-mediated proteolysis pathway-related genes are shown in *Figure 7—figure supplement 1b*. The most significantly upregulated mRNAs including UBA3, UBE2E1, and HERC3 were listed in *Supplementary file 3*.

Therefore, it can be speculated that the ER stress response (ERSR) enabled a proportion of STS26T cells to survive the NIR-III laser treatment through improving protein-folding capacity and ubiquitylating unfolded proteins.

Whole-transcriptome sequencing and corresponding bioinformatics analyses were also performed on S462 cells (*Figure 7—figure supplement 2*). However, the ER pathway was not enriched in the analyses, indicating that heterogeneity exists in response to NIR-III laser treatment among different MPNST cell types. Notably, STS26T cells exhibited stronger resistance to NIR-III laser treatment than S462 cells, as demonstrated by flow cytometric analysis (*Figure 7—figure supplement 3*).

## Discussion

In clinic, the main management for MPNSTs is still surgical resection with sufficient margins, which is often highly risky due to expected damage to adjacent nerves and neurovascular bundles. Therefore,

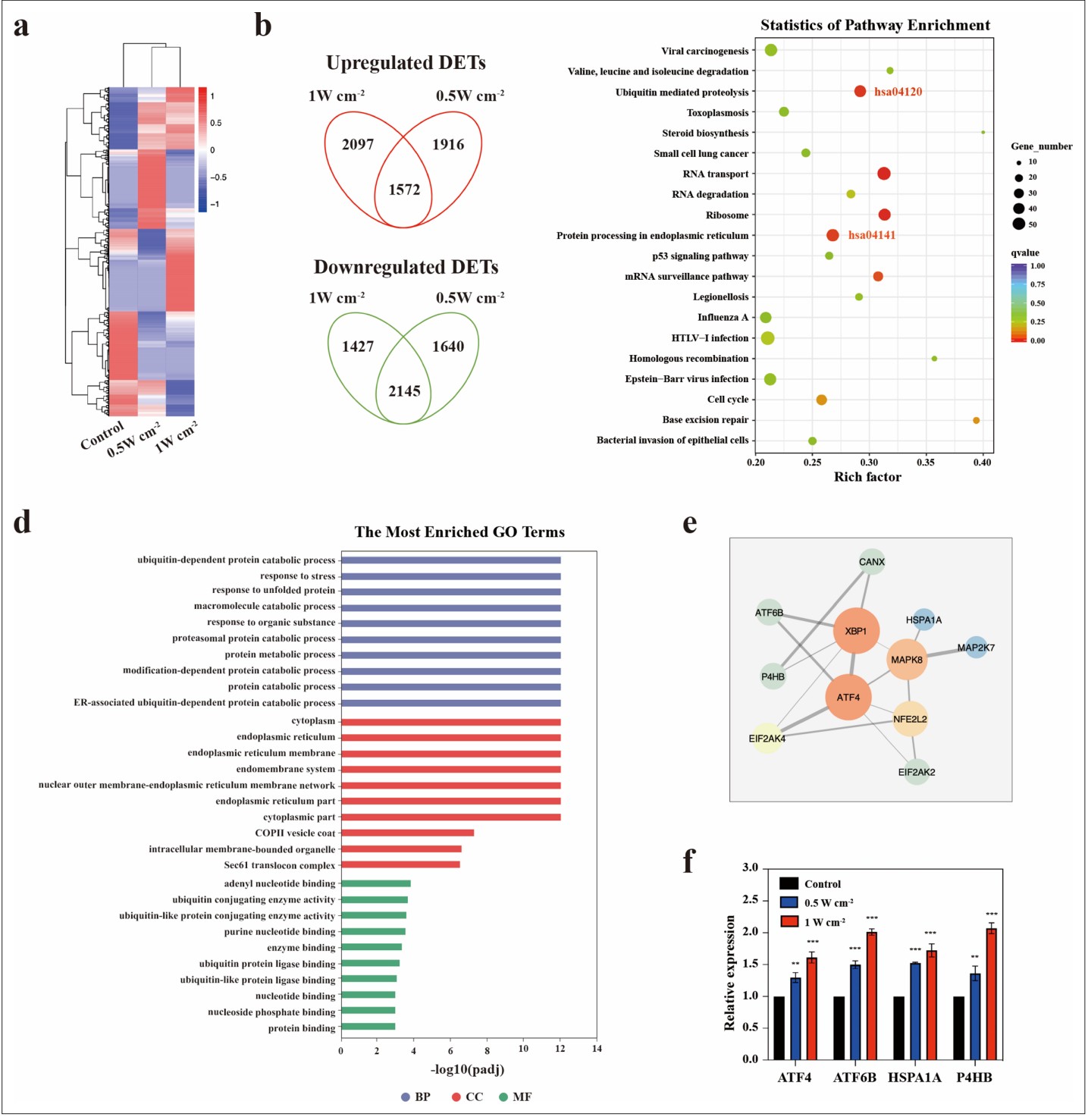

**Figure 7.** Endoplasmic reticulum stress triggered by NIR-III laser treatment. (**a**) Heatmap of the mRNA screening data for STS26T tumors with or without NIR-III laser treatment at the indicated power densities. Blue indicates a low abundance of mRNA, and red indicates a high abundance. (**b**) Venn diagram showing the intersection of differentially expressed transcripts (DETs) in 0.5 and 1 W cm⁻² NIR-III laser treatment group. (**c**) Top 20 pathways revealed by Kyoto Encyclopedia of Genes and Genomes (KEGG) functional enrichment among DETs simultaneously upregulated or downregulated in the NIR-III (0.5 W cm⁻², 5 min) and NIR-III (1 W cm⁻², 5 min) groups. The color indicates the q value; red represents a lower q value, while green represents a higher q value. A lower q value indicates more significant enrichment. The point size indicates the number of DET-related genes. (**d-e**) Gene Ontology (GO) functional enrichment (**d**) and protein-protein interaction network (**e**) of the genes enriched in the protein processing in endoplasmic reticulum pathway. (**f**) mRNA expression levels of ATF4, ATF6B, HSPA1A, and P4HB were measured by RT-qPCR in the indicated groups. All data are presented as mean ± SD and were analyzed with the unpaired t-test (n=3). **p<0.01 and ***p<0.001.

*Figure 7 continued on next page*

*Figure 7 continued*

The online version of this article includes the following figure supplement(s) for figure 7:

**Figure supplement 1.** Alterations in ubiquitin-mediated proteolysis pathway triggered by NIR-III laser treatment.

**Figure supplement 2.** Therapeutic mechanisms exploration on S462 cells by transcriptome high-throughput sequencing.

**Figure supplement 3.** Flow cytometric analysis and the corresponding statistical analysis using Annexin V-FITC/propidium iodide (PI) kits in (**a**) STS26T cells and (**b**) S462 cells seeded in 24-well culture plates after different treatments.

it is applicable to combine preoperative NIR-III photothermal laser treatment and surgery to reduce the scope and the risk of surgery. In addition, postoperative local recurrence and distant metastasis are unfavorable factors affecting the outcomes of MPNST patients. The postoperative application of NIR-III laser treatment at the resection margin could be a potent modality to reduce the risk of local recurrence. However, further experiments and clinical studies are necessary for verification of the safety and effectiveness of these clinical applications.

Mechanically, it is possible that STS26T MPNST cells strongly enhanced the protein processing capacity of the ER to survive the ER stress induced by NIR-III laser treatment. The specific mechanism and regulatory networks need to be explored in the future. In addition, it is speculated that the combination of NIR-III laser treatment with an ER stress inhibitor (such as icariin or ghrelin) could potentially increase therapeutic effectiveness (*Wang et al., 2019*; *Li et al., 2019*), which should be further verified in future studies. Although the tumor ablation effect is influenced by individual differences in molecular response and treatment efficiency, an optimal effect can still be achieved by optimizing the power density and irradiation time of the NIR-III laser.

## Conclusions

In summary, we developed a distinct NIR-III photothermal laser treatment for MPNSTs that does not involve traditional PTAs. The safety, effectiveness, and underlying mechanisms were systematically assessed both in vitro and in vivo, which verified the excellent antineoplastic effect originating from the remarkable photothermal conversion efficiency of the NIR-III laser. This work provides preclinical evidence for the effectiveness of NIR-III treatment against MPNSTs. Further clinical studies are needed to establish safe and optimal therapeutic parameters. It is expected that this PTAs-free NIR-III PTT approach may be further generalized for the treatment of other benign and malignant skin and soft-tissue tumors.

## Acknowledgements

This work was supported by grants from National Natural Science Foundation of China (82102344; 82172228); Shanghai Rising Star Program supported by Science and Technology Commission of Shanghai Municipality (20QA1405600); Natural Science Foundation of Shanghai (22ZR1422300); Science and Technology Commission of Shanghai Municipality (19JC1413); 'Chenguang Program' supported by Shanghai Education Development Foundation (SHEDF) (19CG18); Shanghai Municipal Key Clinical Specialty (shslczdzk00901); Innovative research team of high-level local universities in Shanghai (SSMU-ZDCX20180700). We also acknowledge Biorender for providing us with a platform to create pictures.

## Additional information

### Funding

| Funder | Grant reference number | Author |
| --- | --- | --- |
| National Natural Science Foundation of China | 82102344 | Zhichao Wang |
| National Natural Science Foundation of China | 82172228 | Qingfeng Li |

| Funder | Grant reference number | Author |
| --- | --- | --- |
| Shanghai Rising star Program | 20QA1405600 | Zhichao Wang |
| Natural Science Foundation of Shanghai | 22ZR1422300 | Zhichao Wang |
| Science and Technology Commission of Shanghai Municipality | 19JC1413 | Qingfeng Li |
| Shanghai Education Development Foundation | 19CG18 | Zhichao Wang |
| Shanghai Municipal Key Clinical Specialty | shslczdzk00901 | Qingfeng Li |
| Innovative research team of high-level local universities in Shanghai | SSMU-ZDCX20180700 | Qingfeng Li |

The funders had no role in study design, data collection and interpretation, or the decision to submit the work for publication.

## Author contributions

Yihui Gu, Chengjiang Wei, Meiqi Chang, Data curation, Investigation, Methodology, Writing – original draft; Zhichao Wang, Data curation, Funding acquisition, Investigation, Methodology, Writing – original draft; Yuehua Li, Investigation, Writing - review and editing, Experiments in the revision; Wei Feng, Investigation, Methodology; Wei Wang, Data curation, Investigation; Yu Chen, Conceptualization, Data curation, Supervision, Project administration; Qingfeng Li, Conceptualization, Supervision, Funding acquisition, Investigation

## Author ORCIDs

Yihui Gu  http://orcid.org/0000-0002-9321-1768
Meiqi Chang  http://orcid.org/0000-0003-2211-4044
Qingfeng Li  http://orcid.org/0000-0001-7822-618X

## Ethics

Human subjects: The use of photos and magnetic resonance images of MPNST patients was approved by the Ethics Committee of Shanghai Ninth People's Hospital, Shanghai Jiao Tong University School of Medicine (Reference number: SH9H-2019-T163-2). Informed consent and consent to publish were obtained from patients under institutional review board protocols.

All animals received humane care in compliance with the guidelines outlined in the Guide for the Care and Use of Laboratory Animals. All procedures were performed in accordance with the guidelines approved by the Shanghai Medical Experimental Animal Care Commission (IACUC: 2019-0008).

## Decision letter and Author response

Decision letter https://doi.org/10.7554/eLife.75473.sa1
Author response https://doi.org/10.7554/eLife.75473.sa2

# Additional files

## Supplementary files

- Supplementary file 1. Laser parameters (test after preheating for 5 min).
- Supplementary file 2. Top 10 upregulated genes related to hsa04141.
- Supplementary file 3. Top 10 upregulated genes related to hsa04120.
- Transparent reporting form

## Data availability

The generic database-'Dryad' has been chosen. The unique identifier: doi:10.5061/dryad.3bk3j9km7
All data generated or analysed during this study are included in the manuscript and supporting file.

The following dataset was generated:

| Author(s) | Year | Dataset title | Dataset URL | Database and Identifier |
|---|---|---|---|---|
| Gu Y, Wang Z, Wei C, Feng W, Wang W, Chang M, Chen Y Li Q | 2022 | Photonic hyperthermia of malignant peripheral nerve sheath tumors at the third near-infrared biowindow | https://doi.org/10.5061/dryad.3bk3j9km7 | Dryad Digital Repository, 10.5061/dryad.3bk3j9km7 |

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
