## [Editor Report]

Li et al. report on their effort in NIR-III laser-based photothermal therapy for MPNSTs treatment. Importantly, the biosafety issues of photothermal agents could be circumvented through the introduction of NIR-III laser with relatively high penetration depths and low optical scattering effects. in vitro and in vivo results corroborated the relevant conclusions. This work features high novelty on photonic tumor therapy.

---

## [Decision Letter]

**Decision letter after peer review:**

Thank you for submitting your article "Photonic hyperthermia of malignant peripheral nerve sheath tumors at the third near-infrared biowindow" for consideration by *eLife*. Your article has been reviewed by 3 peer reviewers, including Wei Tao as Guest Editor and Reviewer #1, and the evaluation has been overseen by Wafik El-Deiry as the Senior Editor. The following individual involved in the review of your submission has agreed to reveal their identity: Gongwei Wu (Reviewer #2).

Essential revisions:

1) Please note Reviewer #2's comment on mechanism studies, which could provide more information for future readers – "The mechanism studies in this manuscript provided some information, but a deeper investigation would improve this manuscript."

2) Please note Reviewer #3's comment on comparison, which could highlight the novelty of this study – "The tumor-therapeutic safety and efficacy of NIR-III laser is suggested to be compared with the traditional and mostly used NIR-I and NIR-II lasers in together with photothermal agents."

*Reviewer #1 (Recommendations for the authors):*

Li et al. reported in this manuscript on their effort in NIR-III laser-based photothermal therapy for MPNSTs treatment. Importantly, the biosafety issues of photothermal agents could be circumvented through the introduction of NIR-III laser with relatively high penetration depths and low optical scattering effects. In vitro and in vivo results corroborated the relevant conclusions. This work features high novelty on photonic tumor therapy.

*Reviewer #2 (Recommendations for the authors):*

Malignant peripheral nerve sheath tumors (MPNSTs) are a kind of dismal and relatively rare tumors that account for nearly 5% of the 15,000 soft tissue sarcomas diagnosed in the United States each year. Gu et al., developed a unique near-infrared laser at the third biowindow (NIR-III) photothermal therapy (PTT) method to treat MPNSTs without photothermal agents (PTAs), which eliminated the limitations of PTAs to the clinical practice and showed good efficacies in both some in vitro and in vivo MPNSTs models. Overall, the authors developed and applied a unique NIR-III PPT to treat MPNSTs and demonstrated its potential to translate it for the treatment of patients with MPNSTs.

Some comments to strengthen this manuscript.

1. It is great and the data is solid that the authors compared NIR-I, NIR-II and NIR-III, and showed NIR-III is suitable for the treatment of MPNSTs. A future study would be the optimization of the best biowindow/laser among NIR-III which would improve this manuscript and be good for clinical practice.

2. The authors examined some major organs to test the biosafety of the treatment, however, the authors didn't examine the tissues near the tumor and the treatment spot, such as the tissues beside and behind the tumor/treatment spot.

3. The mechanism studies in this manuscript provided some information, but a deeper investigation would improve this manuscript.

Some detailed comments.

1. This manuscript is somehow rough and lacks much detailed information, the authors should carefully revise the manuscript and provide that information to the readers. For example, how many S462 cells were subQ injected, which time point/when the tumor samples were collected for RNAseq, and how many biological replicates, the authors should add the biological replicates number for each experiment in all figure legends, how did Figure 3b and d were performed, and many and so on.

2. The authors should provide more details about the RNAseq datasets and how did they analyze the data. Current analysis showed that more than 11,000 genes were up-regulated and more than 8,000 genes were down-regulated after NIR-III treatment with either 1 W cm-2 or 0.5 W cm-2, which means around 20,000 genes were significantly regulated, that's almost all of the genes were regulated since human just express around 20,000 genes. The qPCR result in Figure 6d showed that ATF4 only regulated to 1.2 times compared to the control after 0.5 W cm-2, which is obviously not significantly regulated. Why did authors choose ATF4, ATF6B, HSPA1A and P4HB for validation, why not XBP1, MAPK8 or other genes? A deeper analysis should be performed and a detailed description of the samples is also helpful. The authors also need to deposit the RNAseq to a public database.

3. In Figure 3e, the cell morphology of the two Controls are very different, why?

4. It's better to add the data of 0.25W for Figure 3e and f, which have fewer dead cells.

5. Figure 4g is an IHC result instead of an IF result, not consistent with what the authors described in the manuscript. The 1 W cm-2 and 3.0 mm conditions look like having more positive staining cells of caspase-3 than 1 W cm-2 1.5 mm condition. It's better to have a statistics analysis here.

6. Figure 6f should be re-format, current formation missed some legend.

7. For Figure 5a, the control curve should be stopped at day 7.

8. Which sample was chosen to be represented here in Figure 5g?

9. The author should re-gate the flow data for Figure S6a, current gating is not correct.

Statistical analysis is necessary for the flow data.

10. For Figure 3a-d, the control results should be included.

*Reviewer #3 (Recommendations for the authors):*

The premise behind this manuscript is important and timely both for physician scientists and clinicians. The present study provides an intriguing noninvasive therapy for malignant peripheral nerve sheath tumors that accelerates the clinical translation of photonic therapy while avoiding the biocompatibility issues arising from photothermal agents. Gu et al. reported and developed a unique near-infrared laser at the third biowindow (NIR III) for photothermal treatment of malignant peripheral nerve sheath tumors (MPNSTs). Unlike the traditional photothermal therapy involving organic or inorganic photothermal agents, this study explored a new strategy employing laser in safe power density to treat cancer in high efficiency, which is interesting and promising in clinical transformation. I think this study is well organized. The result and conclusion are supported by the given data.

Therefore, I recommend it to publish in eLife after addressing the following issues.

1. The front size in figures such as Figure 2b-I should be magnified. The fluorescent intensity in Figures 3e, 3f and 5g would be better quantified.

2. The cell size in the control group for 2 min treatment appears bigger than for 5 min treatment in Figure 3e. Please check the scale bar size in Figure 3e. I am also confused about the statistic analysis in Figures 5a, 5b and 5e. Please check and correct them.

3. The black fame in Figures 4f-h and 5g should be unified. There is a spacing between Time and (s) in Figures 4a and 4c, 100 and μm in Figures 4f-h.

4. The scale bar in Figures 4h and 5g should be clearer. Furthermore, 50 μm not 50 um in Figure 5g. Please correct it. According to the requirements of eLife, the format of reference should be unified.

5. The tumor-therapeutic safety and efficacy of NIR-III laser are suggested to be compared with the traditional and mostly used NIR-I and NIR-II lasers together with photothermal agents.

[Editors’ note: further revisions were suggested prior to acceptance, as described below.]

Thank you for resubmitting your work entitled "Photonic hyperthermia of malignant peripheral nerve sheath tumors at the third near-infrared biowindow" for further consideration by *eLife*. Your revised article has been evaluated by Wafik El-Deiry (Senior Editor) and a Guest Editor.

The manuscript has been improved but there are some remaining issues that need to be addressed, as outlined below:

Please carefully address the comments from Reviewer #2.

*Reviewer #2 (Recommendations for the authors):*

The revised manuscript is substantially improved. The authors answered most of my questions and comments. My last concern is still about the RNA-seq datasets. The authors provided the details in the revised manuscript that the authors only performed 1 replicate for each group and used the 'edgeR' package for differential expression analysis. 'edgeR' package requires multiple replicates instead of only 1, and this may be the reason that there were around 20,000 genes were significantly regulated and RT-qPCR couldn't perfectly validate the RNA-seq result. I would like to suggest performing more replicates for RNA-seq or deleting the data.

*Reviewer #3 (Recommendations for the authors):*

The authors have fully addressed my concerns. I think it can be accepted now.

---

## [Author Response]

Reviewer #1 (Recommendations for the authors):Li et al. reported in this manuscript on their effort in NIR-III laser-based photothermal therapy for MPNSTs treatment. Importantly, the biosafety issues of photothermal agents could be circumvented through the introduction of NIR-III laser with relatively high penetration depths and low optical scattering effects. In vitro and in vivo results corroborated the relevant conclusions. This work features high novelty on photonic tumor therapy.

Thank you very much for your constructive suggestion and kind recommendation. Please find the following detailed responses.

Reviewer #2 (Recommendations for the authors):Malignant peripheral nerve sheath tumors (MPNSTs) are a kind of dismal and relatively rare tumors that account for nearly 5% of the 15,000 soft tissue sarcomas diagnosed in the United States each year. Gu et al., developed a unique near-infrared laser at the third biowindow (NIR-III) photothermal therapy (PTT) method to treat MPNSTs without photothermal agents (PTAs), which eliminated the limitations of PTAs to the clinical practice and showed good efficacies in both some in vitro and in vivo MPNSTs models. Overall, the authors developed and applied a unique NIR-III PPT to treat MPNSTs and demonstrated its potential to translate it for the treatment of patients with MPNSTs.Some comments to strengthen this manuscript.1. It is great and the data is solid that the authors compared NIR-I, NIR-II and NIR-III, and showed NIR-III is suitable for the treatment of MPNSTs. A future study would be the optimization of the best biowindow/laser among NIR-III which would improve this manuscript and be good for clinical practice.

Thank you very much for your positive comment and constructive suggestions, which are highly appreciated. Please find the following point-to-point responses to your comments and suggestions.

2. The authors examined some major organs to test the biosafety of the treatment, however, the authors didn't examine the tissues near the tumor and the treatment spot, such as the tissues beside and behind the tumor/treatment spot.

Thanks very much for your valuable question, which is highly appreciated. The photos in Figure 5c documented the alteration of the skin beside the treatment spot, which demonstrated that the scars caused by NIR-III treatment was completely healed after 2 weeks. Moreover, no dyskinesia and behavioral abnormalities were observed after the treatment, indicating the integrity of bones and muscles in all mice. The organs behind the treatment spot include the kidneys, liver, and intestines. No significant histological alteration was observed in kidney and liver according to H&E staining. In addition, all mice demonstrated negligible weight fluctuations after the treatment, indicating that the function of intestine was not affected.

3. The mechanism studies in this manuscript provided some information, but a deeper investigation would improve this manuscript.

Thank you very much for your kind suggestion. Please find the following detailed responses.

Some detailed comments.1. This manuscript is somehow rough and lacks much detailed information, the authors should carefully revise the manuscript and provide that information to the readers. For example, how many S462 cells were subQ injected, which time point/when the tumor samples were collected for RNAseq, and how many biological replicates, the authors should add the biological replicates number for each experiment in all figure legends, how did Figure 3b and d were performed, and many and so on.

We appreciate your valuable suggestion. We are sorry for the inconvenience caused by our carelessness, the relevant important information have been added and the manuscript has been revised carefully, which have been highlighted in yellow for your convenience. We sincerely hope that the revised part now meets your expectation.

2. The authors should provide more details about the RNAseq datasets and how did they analyze the data. Current analysis showed that more than 11,000 genes were up-regulated and more than 8,000 genes were down-regulated after NIR-III treatment with either 1 W cm-2 or 0.5 W cm-2, which means around 20,000 genes were significantly regulated, that's almost all of the genes were regulated since human just express around 20,000 genes. The qPCR result in Figure 6d showed that ATF4 only regulated to 1.2 times compared to the control after 0.5 W cm-2, which is obviously not significantly regulated. Why did authors choose ATF4, ATF6B, HSPA1A and P4HB for validation, why not XBP1, MAPK8 or other genes? A deeper analysis should be performed and a detailed description of the samples is also helpful. The authors also need to deposit the RNAseq to a public database.

Thank you very much for pointing out these issues. We performed the differential expression analysis at transcript level and a corrected P-value of 0.05 was set as the thresholds for significant differential expression. Despite of the quantity of the differentially expressed transcripts (DETs), several DETs correspond to the same gene. We have revised the relevant content in the manuscript.

According to the RNA-seq, the transcript ENST00000404241 encoding ATF4 was significantly up-regulated in the NIR (0.5 W cm^-2^) group (logFC = 2.29). However, the PCR results showed that ATF4 expression level was only up-regulated 1.2-fold at the transcriptional level. We believed the reason was that PCR detected the expression of all ATF4 transcripts (including ENST00000337304, ENST00000404241 and ENST00000396680), thus resulting in a lower fold upregulation of ATF4. In addition, considering the efficiency of transcription factors, which means very small amounts of transcription factors could exert transcriptional regulation, it is believed that the upregulation of ATF4 was biologically meaningful.

According to previous investigations and the RNA-seq results, it can be speculated that ATF4 and ATF6B are the key transcription factors that mediate unfolded protein response (UPR) in STS26T xenograft (Annu Rev Pathol.2015;10:173-94). Therefore, we first verified the upregulation of these transcription factors using RT-qPCR method. Furthermore, ATF4 and ATF6B could subsequently upregulate the expression of ER charperones, which was critical for UPR. Meanwhile, the upregulation of the top upregulated ER chaperones genes P4HB and HSP1A1 using RT-qPCR approach has been further confirmed.

According to your kind suggestion, we have further supplemented the analysis related to the ubiquitin mediated proteolysis pathway and refined the manuscript. In addition, we have supplemented the method of RNA-seq and added the data availability statement: The datasets generated for this study can be found in DRYAD at https://doi.org/10.5061/dryad.3bk3j9km7. All the changes made in the revised manuscript have been highlighted in yellow for your convenience. Thank you very much again for pointing out this issue.

3. In Figure 3e, the cell morphology of the two Controls are very different, why?

Thanks very much for your constructive question. After in-depth analysis, it can be concluded that the main reasons for the difference in morphology are the unfixed nature of STS26T cells and the discrepancy in the imaging waiting time. Based on this point, the related experiments have been re-work to minimize the impact of the above-mentioned problems. We have updated Figure 3e in the revised manuscript.

4. It's better to add the data of 0.25W for Figure 3e and f, which have fewer dead cells.

Thank you very much for your kind reminding, which is highly appreciated. We have added the data of 0.25W in Figure 5e and f in the revised manuscript according to your kind suggestion.

5. Figure 4g is an IHC result instead of an IF result, not consistent with what the authors described in the manuscript. The 1 W cm-2 and 3.0 mm conditions look like having more positive staining cells of caspase-3 than 1 W cm-2 1.5 mm condition. It's better to have a statistics analysis here.

Thanks very much for your constructive question. We are sorry for the inconvenience caused by our carelessness and thank you very much for pointing out this issue. The description of “immunofluorescence staining” has been replaced by “immunohistochemical staining”. And the statistic analysis of anti-Cleaved Caspase-3-positive cells and Ki67-positive cells have been added into revised supporting information (Figure 4—figure supplement 1). In addition, we modified Figure 4g with the more representative images.

6. Figure 6f should be re-format, current formation missed some legend.

We are sorry for the inconvenience caused by our carelessness and thank you very much for pointing out this issue. We have updated Figure 6f in the revised manuscript according to your kind suggestion.

7. For Figure 5a, the control curve should be stopped at day 7.

Thank you very much for your kind reminding, which is highly appreciated. We have updated Figure 5a in the revised manuscript.

8. Which sample was chosen to be represented here in Figure 5g?

Thanks very much for your constructive question. We have added the information about the sample in Figure 5g in the “Materials and methods- in vivo PTT” section.

9. The author should re-gate the flow data for Figure S6a, current gating is not correct.Statistical analysis is necessary for the flow data.

Thanks very much for your constructive and professional question, which is highly appreciated. The flow data for Figure 6—figure supplement 3 has been reprocessed. The corresponding statistical analysis have been added into Figure 6—figure supplement 3. We have updated the related figure in the revised supporting information according to your kind suggestion.

10. For Figure 3a-d, the control results should be included.

Thanks very much for your constructive and professional question, which is highly appreciated. We have added the control group into Figure 3a-d according to your valuable suggestion.

Reviewer #3 (Recommendations for the authors):The premise behind this manuscript is important and timely both for physician scientists and clinicians. The present study provides an intriguing noninvasive therapy for malignant peripheral nerve sheath tumors that accelerates the clinical translation of photonic therapy while avoiding the biocompatibility issues arising from photothermal agents. Gu et al. reported and developed a unique near-infrared laser at the third biowindow (NIR III) for photothermal treatment of malignant peripheral nerve sheath tumors (MPNSTs). Unlike the traditional photothermal therapy involving organic or inorganic photothermal agents, this study explored a new strategy employing laser in safe power density to treat cancer in high efficiency, which is interesting and promising in clinical transformation. I think this study is well organized. The result and conclusion are supported by the given data.

Thank you very much for your positive comment and constructive suggestions, which are highly appreciated. Please find the following point-to-point responses to your comments and suggestions.

Therefore, I recommend it to publish in eLife after addressing the following issues.1. The front size in figures such as Figure 2b-I should be magnified. The fluorescent intensity in Figures 3e, 3f and 5g would be better quantified.

Thanks very much for your kind reminding, which is highly appreciated. The font sizes in figures have been magnified according to your valuable suggestion in the revised manuscript. The fluorescent intensity in Fig 3e, 3f and 5g have been quantified and displayed in Fig. 3-figure supplement 1 and Fig. 5-figure supplement 1.

2. The cell size in the control group for 2 min treatment appears bigger than for 5 min treatment in Figure 3e. Please check the scale bar size in Figure 3e. I am also confused about the statistic analysis in Figures 5a, 5b and 5e. Please check and correct them.

Thanks very much for your constructive and professional question. It can be concluded that the main reasons for the difference in morphology and cell size are the unfixed nature of STS26T cells and the discrepancy in the imaging waiting time. Based on this point, the related experiments have been re-work to minimize the impact of the above-mentioned problems. We have updated Fig 3e in the revised manuscript. According to your valuable suggestion, the statistic analysis in Figures 5a, 5d and 5e have been renewed.

3. The black fame in Figures 4f-h and 5g should be unified. There is a spacing between Time and (s) in Figures 4a and 4c, 100 and μm in Figures 4f-h.

We are sorry for the inconvenience caused by our carelessness and thank you very much for pointing out these issues. The related questions about the black fame and spacing have been solved in the revised manuscript.

4. The scale bar in Figures 4h and 5g should be clearer. Furthermore, 50 μm not 50 um in Figure 5g. Please correct it. According to the requirements of eLife, the format of reference should be unified.

We are sorry for the inconvenience caused by our carelessness. The scale bar in Figures 4h and 5g has been revised, and the spelling mistake about 50 μm in Figure 5g has been corrected. The format of reference has been unified.

5. The tumor-therapeutic safety and efficacy of NIR-III laser are suggested to be compared with the traditional and mostly used NIR-I and NIR-II lasers together with photothermal agents.

Thanks very much for your constructive question. In terms of tumor-therapeutic safety, the traditional NIR-I or NIR-II photothermal therapy exist inherent biosafety issues due to the introduction of photothermal agents. Inorganic photothermal agents have intrinsic issues such as poor biodegradability, disappointing processability and apparent cytotoxicity. Although the advantageous properties of organic agents include distinct biodegradability and biocompatibility, the deficient photothermal conversion efficiency and photothermal stability limit these further biological applications. However, NIR-I or NIR-II laser irradiation alone can not reach the appropriate temperature to kill tumors. Therefore, NIR-III laser treatment can circumvent the biosafety problems of photothermal agents. In terms of tumor-therapeutic efficacy, NIR-III lasers possess relatively high penetration depths and low optical scattering effects compared with NIR-I and NIR-II lasers. The total attenuation lengths of human prostate and breast cancer have been revealed to be higher in the NIR-III optical windows than in the other NIR regions, confirming the potential for application of NIR-III lasers in tumor therapy (Sordillo D C, Sordillo L A, Sordillo P P, et al. Fourth near-infrared optical window for assessment of bone and other tissues[C]//Photonic Therapeutics and Diagnostics XII. SPIE, 2016, 9689: 499-506.).

Finally, we greatly appreciate and thank the reviewers’ kind, professional and constructive reminding, comments and suggestions for this manuscript. We have tried our best to address all these issues as possible as we can. We sincerely hope that the revised manuscript has addressed all the comments and suggestions as kindly raised by the reviewers and meet the publication standard of eLife. Thank you very much.

[Editors’ note: further revisions were suggested prior to acceptance, as described below.]

The manuscript has been improved but there are some remaining issues that need to be addressed, as outlined below:Please carefully address the comments from Reviewer #2.Reviewer #2 (Recommendations for the authors):The revised manuscript is substantially improved. The authors answered most of my questions and comments. My last concern is still about the RNA-seq datasets. The authors provided the details in the revised manuscript that the authors only performed 1 replicate for each group and used the 'edgeR' package for differential expression analysis. 'edgeR' package requires multiple replicates instead of only 1, and this may be the reason that there were around 20,000 genes were significantly regulated and RT-qPCR couldn't perfectly validate the RNA-seq result. I would like to suggest performing more replicates for RNA-seq or deleting the data.

Thank you very much for your constructive suggestion and kind recommendation. We also agree that biological replicates can provide more solid results. Therefore, we planned to perform more technical replicates using the same RNA samples returned from sequencing. We performed quality control checks on the RNA samples. In addition, considering the updates of bioinformatics analysis process in the sequencing company, we re-analyzed the original data and compared the results. However, our plan to perform technical replicates failed due to the partial degradation of the RNA samples and the impact of the alterations in the bioinformatics analysis process on the results. Due to the failure of the plan, we can only redesign and reperform the relevant experiments (including in vivo modeling, NIR treatment, sampling, sequencing, and bioinformatics analysis). However, considering the long duration of the experiment, we temporarily gave up on this attempt.

We still kept the results of RNA-seq in the manuscript for the following reasons: 1. the results of RNA-seq are relatively stable across different biological replicates. 2. Many previous investigations also confirmed the reliability of RNA-seq data without replicates. (Adv. Funct. Mater. 2020, 30, 2002610. DOI:10.1002/adfm.202002610; Cancer Cell. 2018 Mar 12;33(3):368-385.e7.) 3. In this study, RNA-seq was performed on tumors from the control, NIR-III (0.5 W cm^-2^, 5 min), and NIR-III (1 W cm^-2^, 5 min) groups. We performed functional enrichment analysis of the differentially expressed transcripts (DETs) in the two treatment groups respectively. The results of the two groups were similar and consistent with those in the manuscript. Alterations in protein processing in the endoplasmic reticulum and ubiquitin-mediated proteolysis pathways were both significant in the two treatment groups. (In the manuscript, the enrichment analysis was carried out on the common DETs of the two treatment groups).

**Author response table 1. sa2table1:** STS26T NIR-III 1 W cm-2.

Pathway term	qvalue	Gene number
Protein processing in endoplasmic reticulum	0.03501825	68
Ubiquitin mediated proteolysis	0.01292571	62

**Author response table 2. sa2table2:** STS26T NIR-III 0. 5 W cm-2

Pathway term	qvalue	Gene number
Protein processing in endoplasmic reticulum	0.02231676	55
Ubiquitin mediated proteolysis	0.00044313	56

Regarding the use of the 'edgeR' package for differential expression analysis, we reviewed relevant literature and discussed the issue with technical support. For samples without biological replicates, differential expression analysis of two conditions was performed using the edgeR R package (3.12.1). According to the edgeR User's Guide, differential expression analysis using edgeR R package can be performed in samples with and without replicates. edgeR proposed 4 methods for the differential expression analysis in samples without replicates. (edgeR: differential analysis of sequence read count data. User’s Guide. Chen, Y. et al. (2022))

Regarding the number of differentially expressed transcripts (DETs), our previous response was that a gene may correspond to several transcripts, resulting in a large number of transcripts dysregulated. In addition, we need to state that there are many low-abundance transcripts (transcripts with low FPKM values) in our DET list. Most transcripts, especially mRNAs, with low FPKM values are not biologically significant. However, there is no standard definition of low-abundance transcripts/genes, so the treatment of these transcripts/genes varies across studies. We summarized the common treatment methods of these transcripts/genes: 1. keep all the differentially expressed genes for subsequent functional analysis. 2. perform low-expression gene filtering before differential analysis. 3. Change the threshold of the DETs in differential analysis (eg. Padj<0.05, |LFC|>1, FPKM>1) in differential analysis. 4. No alterations in differential expression analysis and only transcripts with sufficient expression are selected for subsequent functional analysis. We used method 4 in the previous manuscript (We selected DETs with FPKM>1 for enrichment analysis).

Your valuable comments made us realize that genes with low expression should be deleted in the differential expression analysis as they are not biologically meaningful, and it may be confusing to readers and other researchers. Therefore, after reviewing relevant literature and consulting bioinformatics technical support, we changed the threshold of DETs in differential analysis to padj <0.05, |logFC| >1, and FPKMmax >1. (FPKMmax stands for the maximum expression level of a transcript from the two samples compared). (Cancer Res (2019) 79 (19): 4840–4854.; Genome Biol. 2014 Aug 13;15(8):429.; Front Cell Dev Biol. 2020 Dec 10;8:599494.) We reperformed subsequent analyses based on the new DET list, and the results of the Venn analysis, KEGG, and GO enrichment were updated in the manuscript and the supplementary material. The protein-protein interaction network and the top 10 upregulated genes related to hsa04141/hsa04120 were not modified because none of the DETs were deleted. The RNA-seq analysis method was also modified.

The main objective of this study was to evaluate the efficacy and safety of NIR-III laser treatment in MPNST, so the explorations of the underlying mechanisms of NIR-III treatment effects were preliminary and based only on RNA-seq. We have added a statement of this limitation to the manuscript. We also plan to explore the relevant mechanisms and combination therapy in further investigation.

All the changes made in the revised manuscript have been highlighted in yellow for your convenience.

Finally, we greatly appreciate and thank the reviewers’ kind, professional and constructive reminding, comments and suggestions for this manuscript. We have tried our best to address all these issues as possible as we can. We sincerely hope that the revised manuscript has addressed all the comments and suggestions as kindly raised by the reviewers and meet the publication standard of *eLife*. Thank you very much.